# Therapeutic Role of Carotenoids in Blood Cancer: Mechanistic Insights and Therapeutic Potential

**DOI:** 10.3390/nu14091949

**Published:** 2022-05-06

**Authors:** Yaseen Hussain, Khalaf F. Alsharif, Michael Aschner, Abdulrahman Theyab, Fazlullah Khan, Luciano Saso, Haroon Khan

**Affiliations:** 1Lab of Controlled Release and Drug Delivery System, College of Pharmaceutical Sciences, Soochow University, Suzhou 215000, China; pharmycc@gmail.com; 2Department of Pharmacy, Bashir Institute of Health Sciences, Bharakahu, Islamabad 44000, Pakistan; 3Department of Pharmacy, University of Malakand, Chakdara 18800, Pakistan; abdullah@uom.edu.pk; 4Department of Clinical Laboratory, College of Applied Medical Science, Taif University, P.O. Box 11099, Taif 21944, Saudi Arabia; alsharif@tu.edu.sa; 5Department of Molecular Pharmacology, Albert Einstein College of Medicine, Bronx, NY 10463, USA; michael.aschner@einsteinmed.org; 6Department of Laboratory and Blood Bank, Security Forces Hospital, P.O. Box 14799, Mecca 21955, Saudi Arabia; aalshehri@sfhm.med.sa; 7College of Medicine, Al-Faisal University, P.O. Box 50927, Riyadh 11533, Saudi Arabia; 8Faculty of Pharmacy, Capital University of Science & Technology, Islamabad 44000, Pakistan; fazlullah.Khan@cust.edu.pk; 9Department of Physiology and Pharmacology “Vittorio Erspamer”, Sapienza University, 00185 Rome, Italy; luciano.saso@uniroma1.it; 10Department of Pharmacy, Abdul Wali Khan University Mardan, Mardan 23200, Pakistan

**Keywords:** carotenoids, blood cancer, challenges, carotenoids in blood cancer, therapeutic potentials

## Abstract

Blood cancers are characterized by pathological disorders causing uncontrolled hematological cell division. Various strategies were previously explored for the treatment of blood cancers, including chemotherapy, Car-T therapy, targeting chimeric antigen receptors, and platelets therapy. However, all these therapies pose serious challenges that limit their use in blood cancer therapy, such as poor metabolism. Furthermore, the solubility and stability of anticancer drugs limit efficacy and bio-distribution and cause toxicity. The isolation and purification of natural killer cells during Car-T cell therapy is a major challenge. To cope with these challenges, treatment strategies from phyto-medicine scaffolds have been evaluated for blood cancer treatments. Carotenoids represent a versatile class of phytochemical that offer therapeutic efficacy in the treatment of cancer, and specifically blood cancer. Carotenoids, through various signaling pathways and mechanisms, such as the activation of AMPK, expression of autophagy biochemical markers (p62/LC3-II), activation of Keap1-Nrf2/EpRE/ARE signaaling pathway, nuclear factor kappa-light-chain-enhancer of activated B cells (NF-κB), increased level of reactive oxygen species, cleaved poly (ADP-ribose) polymerase (c-PARP), c-caspase-3, -7, decreased level of Bcl-xL, cycle arrest at the G0/G1 phase, and decreasing STAT3 expression results in apoptosis induction and inhibition of cancer cell proliferation. This review article focuses the therapeutic potential of carotenoids in blood cancers, addressing various mechanisms and signaling pathways that mediate their therapeutic efficacy.

## 1. Introduction

Blood cancer is a group of disorders causing uncontrolled hematological cell division. Blood cancer classification into leukemia, myeloma and lymphoma, is based on the type of blood cell affected [1]. Leukemia is characterized by excessive and abnormal multiplication of leukocytes. It is sub-divided into acute and chronic, and includes acute lymphocytic leukemia, acute myeloid leukemia, chronic lymphocytic leukemia, and chronic myeloid leukemia. Lymphoma is characterized by uncontrolled and abnormal lymphocyte production. The malignant lymphocytes travel through lymphatic system and blood to various organs where they accumulate to form tumors. Lymphoma is sub-divided into Hodgkin’s and Non-Hodgkin’s lymphoma based on the presence or absence of Reed-Sternberg cells, respectively. Myeloma is characterized by uncontrolled multiplication of plasma cells in the bone marrow. It is sub-divided into hyperdiploid and non-hyperdiploid myeloma based on whether the cancerous myeloma has more chromosomes or fewer chromosomes than normal, respectively. Non-hyperdiploid myeloma is more aggressive [1].

Leukemia is a common malignant disorder affecting millions of people across the globe. It is the 15th most commonly diagnosed cancer and the 11th cause of death due to malignancies [2]. Leukemia is universally encountered across the globe having higher prevalence in more developed countries. It has been estimated by the American Cancer Society that in the year 2020, that 178,520 individuals were to be diagnosed with leukemia, myeloma, and lymphoma in the United States (USA). In the USA, the prevalence and mortality of leukemia in males is higher than females [2]. Leukemia is malignant clonal disorder of the organs that produced the blood and involve one or more cell lines in the hematopoietic system. There is diffuse replacement of bone marrow by immature and undifferentiated hematopoietic cells with consequent reduction in the RBCs and thrombocytes in peripheral blood. Leukemia is classified into lymphoid, myeloid and mixed leukemia based on hematopoietic organs involved [3].

## 2. Carotenoids: An Overview

Carotenoids are a large group of plant secondary metabolites containing more than 1000 compounds. It is based on C-40 tetraterpenoids; however, C-30 and C-50 based carotenoids found in bacteria have been recently included in the group [4,5,6]. When two C20 geranyl-geranyldiphosphate molecules are linked together, they produce carotenoids. All carotenoid compounds contain a polyisoprenoid structure, a long conjugated chain of double bonds and an almost bilateral symmetry around the central double bond [7]. Carotenoids have been found in plants, animals, fungi, and bacteria. Several carotenoids are orange, red or yellow color pigments associated with chlorophylls in plants helping against photo-oxidation and enhancing photosynthesis [8]. Carotenoids are divided into two main classes based on the presence (xanthophylls) or absence of oxygen (carotenes). Examples of xanthophylls are zeaxanthin, fucoxanthin and neoxanthin while α- and β-carotene and lycopene belong to the carotene group [9]. These can also be classified into provitamin A and non-provitamin A groups. The former contain α- and β-carotene, and β-cryptoxanthin while the examples of the latter are lutein and zeaxanthin [10]. Other compounds such as apocarotenoids (e.g., retinoids, vitamin A, β-ionone and α-ionone aromatic volatile compounds) are also derived from carotenoids by oxidative cleavage using carotenoid cleavage dioxygenases [7]. Figure 1 shows the sources and classification of carotenoids [11].

Commonly, carotenoids are hydrophobic compounds having very low water solubility that act in hydrophobic areas of the cell. The attachment of polar functional groups to the polyene chain changes the carotenoids polarity, which not only changes their localization within the biomembranes, but also their interaction with several molecules [12]. Figure 2 shows the structures of a few representative carotenoids.

Humans and animals are unable to synthesize carotenoids, yet they are found in their bodies due to dietary intake of food containing carotenoids. These are fat-soluble compounds absorbed from the intestine. Carotenoids are transported by the chylomicrons from intestinal mucosa to blood via lymphatics for delivery to the tissues. In the plasma, these are then transported by the lipoproteins [13]. Carotenoids are delivered to extra hepatic tissues via the interaction of lipoprotein particles with receptors and the degradation by lipoprotein lipase [13,14,15]. In mammals, facilitated diffusion has been found recently to mediate carotenoids intestinal absorption [16]. Several carotenoids are precursors of vitamin A and many are efficient anticancer and antioxidants agents.

Scientists have shown great interest in carotenoids due to correlation between their blood concentration and reduced incidence of chronic diseases [17]. A meta-analysis recently showed that subjects taking α-carotene, β-carotene, and β-cryptoxanthin had lower mortality [18]. On the basis of these health benefits, a health index has been proposed, that states that people having less than 1 μM of total carotenoids plasma concentration are associated with significantly higher risk for chronic diseases [19].

The major health benefits of carotenoids are due to their antioxidant activity, anti-inflammatory activity, and immune system enhancement [20,21]. During the past two decades, the role of carotenoids in the treatment and prevention of cancer has been studied [22,23]. At the molecular level, carotenoids exert their anticancer activity via modulation of several signaling pathways including cell proliferation, cell cycle progression, metastasis, angiogenesis, and apoptosis [24]. Carotenoids alters metastasis via regulatory mechanisms including HIF-1α, glucose transporter 1 (GLUT-1), uPA, RhoGTPase (RhoA, Rac1, and Cdc42), MMPs, E-cadherin, surface glycoprotein CD44 and CXCR4, non-metastatic protein 23 homlog 1 (Nm23-H1), and TIMPs [24]. The molecular mechanism for the bioactivities of carotenoids is illustrated in Figure 3.

## 3. Cancer and Metastasis

Metastasis is one of the hallmarks of cancer and a major cause of treatment failure [25,26]. Drug resistance to chemotherapy also leads to recurrence of cancer and metastasis. [27]. In about 90% of cancer patients, metastasis is the main cause of death [26]. Although early diagnosis or prevention increases the clinical outcomes, cancer is often unfortunately diagnosed in advanced stages when metastasis has occurred [28,29]. Thus, the early detection of metastatic cancer and proper therapeutic intervention is mandatory in the management of advanced cancer [30]. Metastasis involves local invasion, intravasation, extravasation, survival in circulation, and colonization [31] (Figure 4).

Metastasis of solid tumors can be defined as the appearance of secondary tumors distant from the primary site. In case of leukemia (liquid tumors), widespread disease is diagnosed initially, and the exact location of origin (bone marrow, lymph node or spleen) usually remains unknown. In addition to this ambiguity, leukemic cells possess the intrinsic ability of motility, there is disagreement whether leukemia is a metastatic disease. Despite this controversy, leukemia possesses several properties of metastasizing solid tumor cells. These include derivation from mutant clone that has the disease-initiating ability [32,33,34], spread through a cascade of molecular events involving intravasation, extravasation and colonization of tissues [35,36,37], reproducible and specific patterns of organ involvement [38], adaptation to tissue microenvironment that is different from the primary one 13, 14, and widespread disease progression [39,40].

Metastasis heterogeneity and the difference of secondary tumors from the primary tumor as well as additional changes at epigenetic and genetic level are the other obstacles in the treatment of cancer [26]. Carotenoids exert anticancer activity by metastasis modulation and regulation of several molecules and signaling pathways involved in metastasis including chronic inflammation, tumor microenvironment, genetic and epigenetic factors, extracellular matrix, epithelial-mesenchymal transition (EMT), and cancer stem cells [11]. This review article focuses on the therapeutic potential of carotenoids in blood cancer with emphasis on mechanistic insights.

## 4. Challenges in Blood Cancer Treatment

Chemotherapy is one of the treatment options for blood cancer, but unfortunately, it exerts many toxic effects on various organs, including heart, liver, kidney, and brain, to name a few [41]. Chemotherapy leads mortality and morbidity in cancer patients as the drugs have low specificity, and thus, affects any rapidly dividing cell whether cancerous or healthy. Poor metabolism, solubility, and stability of anticancer drugs pose the challenge of limited efficacy, bio-distribution, and toxicity. Therefore, development of highly specific drug with minimum/no toxicity and better pharmacokinetic properties is imperative in the treatment of blood cancer [42,43,44,45,46,47].

Chimeric antigen receptors have been introduced to natural killer cells that have the potential of causing selective cancer cell cytotoxicity and thus used in cellular immunotherapy [48]. In addition, Chimeric antigen receptor T (CAR-T therapy is a versatile approach in the field of oncology that has shown therapeutic benefits in leukemia and lymphoma [49]. CAR-T cell therapy is used in patients suffering from cancers that are difficult to treat, where chemotherapy is ineffective, or where the cancer returns. This therapy has unprecedented efficacy in B cell malignancy especially in anti-CD19 CAR-T cells for B cell acute lymphoblastic leukemia with up to a 90% complete remission rate [50]. However, certain challenges also exist during treatment of hematological cancers, such as the isolation and purification of natural killer cells is a big challenge. The transduction of primary natural killer cells triggers heterogeneous population with decreased expansion. In association, such transduction is complicated as well [51]. Moreover, the structural design of CARs should have an impact on Natural Killer (NK) cells to prevent the hindrance in the clinical application of CAR-NK treatment. Binding with antigens, CAR-NK cell activation and the immune synapses optimal formation are the factors affected by CAR-binding epitope position and the distance from the CAR-NK cell surface [52].

Platelets targeting in blood cancer is another big challenge in cancer therapy because the interlinking of specific platelet with particular hematological malignancy is still unknown [53]. In addition, the use of antiplatelet agents for cancer treatment is hampered by the lack of understanding of conditions involved in eliciting anti-inflammatory, anti-adhesive and anti- angiogenic and anti-mitogenic functions of platelets. The inflammatory process in physiological wound healing is mediated in a natural way which, in turn, switches ON/OFF the epithelial—mesenchymal transition and cell proliferation as well [54]. However, such regulation is abnormal in cancer and a pathogenic chronic wound-healing process is induced. In such scenario, the interaction of platelets with tumor cells for initiation of epithelial—mesenchymal transition remains unknown and poses a continued challenge in blood oncology [55]. It is noteworthy that pharmacological and genetic targeting of platelets might be a suitable strategy for mimicking T-cell therapy, providing opportunities for cancer immunotherapy optimization through inhibition of immune checkpoints molecule and platelets simultaneously [56]. 

Acute myeloid leukemia (AML) often relapses due to leukemic stem cells, which re-initiate malignancy, and are resistant to drugs [57]. In order to address this problem, patients undergo allogenic stem cell transplant. However, this method also has significant mortality and morbidity [58]. The diagnosis of myeloma is confirmed by bone marrow sampling. Young individuals can better tolerate biopsy but it is a challenging job in weak and geriatric patients with bleeding risk and underlying osteoporosis [59].

Prognosis of adult patients with acute lymphoblastic leukemia (ALL) and the treatment regimen carrying several toxicity risks [60,61]. Improvement of prognosis of such patients is a challenging task [62]. Hematopoietic stem cell transplantation is a method of treatment for such patients but this treatment has some difficulties including the donor source, multidrug resistance before and after transplant and the development of graft versus host disease [63,64,65,66]. Cancer immunotherapy is dominated by immune cells, but there are many challenges to its application including production cost, preservation, and transport. Extracellular vesicle (EVs); nano-scaled vesicles naturally secreted by NK cells and many other cells are used to attack tumor cells, but the mechanism cell demise has yet to be clarified [48].

## 5. Carotenoids in Cancer

Given their dietary origin, phytochemicals are presumed as a safer treatment choice. Such dietary based phytochemicals are used to cope with various diseases including cancer [20]. Among these safer phytochemicals, carotenoids belong to the class of tetraterpenes that exhibit C_40_H_56_ hydrocarbon skeleton followed by alternate single and double bond conjugation [67]. Carotenoids impart chemical reactivity and photochemical properties to plants [68]. The presence of lipophilic and hydrophobic compounds in the carotenoid structure pose it low water soluble class of phytochemicals [69]. Carotenoids are mainly classified into carotenes and xanthophylls that give characteristic color in yellow to red spectrum range [70].

For normal physiological processes body cannot synthesize carotenoids and thus have to be obtained from diet [71]. Seven hundred carotenoids have been identified thus far [72], among which 40 are available in human diet [73]. Within human diet the representative carotenoids are lycopene, α/β carotene and xanthophylls such as β-cryptoxanthin, lutein and zeaxanthin [74]. The representative carotenoids in human diet along with their sources are shown in Table 1.

The purified extract of lycopene was evaluated for its anticancer potential in prostate cancer cells. The sera collected from male subjects were investigated and results showed a significant down regulation of p53 and up regulation of BAX/Bcl-2 ratio, resulting in apoptosis [82]. Apart from apoptosis induction, lycopene has the ability to cause cell cycle arrest, suppression of proliferation and modulation of signaling pathways in prostate cancer cells [83]. In a randomized double blind placebo controlled trial lycopene at a dose of 30 mg/day showed a higher concentration in prostate tissue suggesting a clue for loading to prostate cancer tissues [84]. The human prostate cancer cells proliferation was affected by neoxanthin along with apoptosis induction and endogenous antioxidant signals activation [85].

The purified extract of lutein was investigated in MCF-7 breast cancer cell line and the proposed carotenoid significantly inhibited the cell viability. Furthermore, lutein induced apoptosis, suppressed cell survival and antioxidant signals that ultimately resulted in inhibition of breast cancer cell growth [86]. Similarly, lutein in combination with long non-coding RNA cancer susceptibility (CASC9) was evaluated for the antiproliferative activity in MCF-7 breast cancer cells. The proposed hybridized array was evaluated through rtPCR, MTT assay, and dual luciferase reporter studies. Results showed that lutein down regulated the expression of CASC9 and up regulated the expression of miR-590-3p that eventually resulted in inhibition of breast cancer proliferation [87]. Xiaoming et al. reported the significant anti-breast cancer potential of lutein. The mechanism invoked for its anticancer potential is increased production of reactive oxygen species in intracellular environment of triple negative breast cancer. Lutein was also shown to inhibit breast cancer cell growth [88]. The activation of Nrf2/ARE signaling pathway and blocking of NF-κB pathway was found as mechanistic ways for the treatment of human breast cancer through lutein [89]. Astaxanthin through increased reactive oxygen species generation and β-carotene through increased Bax production showed synergistic effect towards the cytotoxic killing of estrogen receptor positive MCF-7 breast cancer cells and estrogen receptor negative breast cancer cells [90,91].

The mechanism and efficacy of lycopene, β-carotene and β-cryptoxanthin was evaluated in smoking-induced oxidative DNA damage in lung cancer cells. The cells were exposed to smoking and treated with proposed carotenoids at different doses. Results showed that carotenoids at high concentrations had prooxidant effects, while at low concentrations showed antioxidant effects [92]. Fucoxanthin carotenoid extracted from Laminaria japonica was investigated for its anticancer potential in lung cancer cells. Transwell and wound healing assays were used for the detection of invasion and migration of lung cancer cells, and western blotting was used for determination of signaling pathways and transitions. Lung metastatic tumor model was used for the in vivo activity of fucoxanthin. Lung cancer cell metastasis was suppressed by fucoxanthin and enhanced the sensitivity of lung cancer cells to Gefitinib [93].

Preclinical research has shown that carotenoids are interesting candidates for the treatment of multiple cancers targeting metastasis, migration, and cell invasion and therefore stand strong candidates for clinical studies in order ascertain their clinical potential. The anticancer potential of carotenoids both in vitro and In vivo is shown briefly in Table 2 along with their mechanism.

## 6. Carotenoids in Blood Cancer

### 6.1. Carotenoids in Leukemia

Chronic lymphocytic leukemia is the most frequent form of blood cancer in adult population [100]. The failure of conventional therapeutic approaches is attributed to the poor response of chronic lymphocytic leukemia to apoptosis resistance and inductive autophagy [101,102]. In this context, recently the effect of carotenoids extracted from pumpkin on leukemia cell line was evaluated. Results of the study showed that compared to untreated cells, the carotenoid extract showed a 40% reduction in cell proliferation without any cytotoxicity. The modulation of autophagy flux and activation of AMPK were the observed mechanisms for delay in cell proliferation. In addition, the expression of autophagy biochemical markers i.e., p62/LC3-II was significantly changed [103]. The anti-blood cancer activity of β-carotenes, such as bixin and capsanthin, was investigated in leukemia K562 cell line. The carotenoids in time- and dose-dependent manner, induced apoptosis, decreased cell viability, and disrupted cell cycle progression. Mechanistically, it was observed that carotenoids inhibited cell proliferation through activation of Keap1-Nrf2/EpRE/ARE signaling pathway [104]. Similarly, using the same cell line, Amir, et. al. investigated the anti-leukemic effect of carotenoid—lycopene, at different dose concentrations ranging from 0 to 100 μg/mL using nano technology scaffold, showing enhanced percentage of apoptotic cells along with significant cell growth inhibition [105]. Another study investigated the inhibition of cell growth in leukemia cell line after administration of carotenoid (β-cryptoxanthin) from nano platform [106].

Among marine carotenoids, heteronemin was explored in cytarabine resistant acute myeloid leukemia cells. The MTT assay results showed that heteronemin significantly sensitized HL-60 cells toward sub toxic doses of cytarabine and showed a synergistic toxicity profile. In association, heteronemin down regulated the activation of activator protein 1 (AP-1), c-myc, mitogen-activated protein kinase (MAPK) and nuclear factor kappa-light-chain-enhancer of activated B cells (NF-κB) showing effective anticancer potential of carotenoid in acute myeloid leukemia [107]. Fucoxanthin isolated from Ishige okamurae induced apoptosis in human leukemia through increased level of reactive oxygen species (ROS), cleaved poly (ADP-ribose) polymerase (c-PARP), c-caspase-3, -7, and decreased level of Bcl-xL at a dosing concentration of 7.5, 15 and 30 µg [108]. Siphonaxanthin is a green algae carotenoid and its antileukemic activity was investigated in human leukemia cell line. Results showed that Siphonaxanthin triggered anti-angiogenic effect via down-regulation of signal transduction by fibroblast growth factor receptor-1. Siphonaxanthin also induced apoptosis with a 2 fold higher cellular uptake as compared to Fucoxanthin [109]. It shows that application of carotenoid is an interesting strategy in coping with the consequences of leukemia.

Chronic myelogenous leukemia is a form of leukemia characterized by over production of white blood cells [110]. The cytotoxic effect of carotenoids extracted from Spirulina platensis was evaluated in chronic myelogenous leukemia cell line (K-562). The cells were treated with various concentrations of *S. platensis* extract i.e., 0.25–50 mg/mL for a period of 72 h. The IC_50_ value was 4.64 mg/mL and significantly comparable with cyclophosphamide [111]. Fucoxanthin’s in vitro cytotoxicity alone and in combination with chemotherapy drugs, doxorubicin and imatinib, was evaluated in human chronic myelogenous leukemia cell lines (TK6 and K562). Doxorubicin decreased cell proliferation and viability in both cell lines; imatinib inhibited cell proliferation in K562 cell line and increased cytotoxicity in TK6 cells while Fucoxanthin decreased cell proliferation in both cell lines with elevated cytotoxicity against K562 cell line [112]. Another study showed cell growth inhibition and apoptosis induction in Jurkat human T-cell leukemia cell line by Crocin, secondary to increased Bax gene expression and inhibition of Bcl-2 [113]. Similarly, in vivo and in vitro studies in human leukemia cell line (HL-60) showed pro apoptotic and antiproliferative effects by Crocin through cell cycle arrest at the G0/G1 phase. In addition, Crocin in xenograft a mouse HL-60 cell line significantly increased Bax expression, reduced tumor weight/size and inhibited Bcl-2 expression [114]. Crocetin showed anti- proliferative effect in promyelocytic leukemia HL-60 cell line [115].

### 6.2. Carotenoids in Lymphoma and Myeloma

Lymphoma is another type of blood cancer that appears in lymphatic system [116]. Findings of a meta-analysis suggested that higher intake of carotenoids, such as α-carotene, β-carotene and lutein, protected against lymphoma [117]. In a recent hospital-based case control study 512 newly diagnosed non-Hodgkin’s lymphoma patients and 512 healthy control group the anti-lymphoma potential of carotenoids was evaluated. Carotenoids intake, such as lycopene, α, and β-carotene was shown to be inversely related to non-Hodgkin’s lymphoma. Furthermore, carotenoids reduced the risk of non-Hodgkin’s lymphoma particularly in smokers [118]. A recent meta-analysis study in 3228 cases of non-Hodgkin’s lymphoma patients showed that α -carotene (RR = 0.87, 95% CI = 0.78–0.97), β-carotene (RR = 0.80, 95% CI = 0.68–0.94) and lutein/zeaxanthin (RR = 0.82, 95% CI = 0.69–0.97) significantly decreased the risk of non-Hodgkin’s lymphoma. Data from preclinical studies confirmed that carotenoids had protective effects against non-Hodgkin’s lymphoma by increasing antioxidant and antiapoptotic activity [22].

Myeloma is blood cancer type that originates from plasma cells of bone marrow [119]. Crocin was shown to significantly reduce the production of reactive oxygen species along with anti-myeloma effect in U266B1 myeloma cell line [120]. Multiple myeloma is associated with abnormal or higher expression of STAT3 [121]. In this context, Raza et al. showed that astaxanthin decreased STAT3 expression leading to inhibition of DU145 cells proliferation [122].

## 7. Discussion

Blood cancer is a group of pathological disorders causing uncontrolled hematological cells division, starting in the bone marrow and spreading throughout the body. Blood cancer is classified into leukemia, myeloma and lymphoma based on the type of blood cell affected. Scientists have shown great interest in carotenoids due to correlation between blood concentration level and reduced incidence of chronic diseases. It has been demonstrated that people with high blood concentration of β-carotene have decreased incidence of all-cause mortality in comparison to those having low blood concentration.

Chemotherapy in blood cancer therapy have posed many adverse consequences and unfortunately failed in blood cancer therapy. Such failure is one among major challenges for researcher working on blood cancer. Chimeric antigen receptors and CAR-T cell therapy have shown therapeutic benefits in blood cancer therapy, however, the isolation, purification, and transduction of natural killer cells for such therapies is quiet challenging. In addition, platelets targeting in blood cancer is another big challenge because the interlinking of specific platelet with particular hematological malignancy remains unknown. Similarly, the introduction of anti–platelet agents for cancer treatment is hampered by the lack of understanding on conditions involved in eliciting of anti-inflammatory, adhesive and anti-angiogenic and mitogenic functions of platelets. The interaction between platelets and tumor cell for triggering epithelial to mesenchymal transition remains unknown. Relapse and drug resistance during blood cancer therapy pose other challenges. Phytochemicals due to the dietary origin are presumed as a safer choice and thanks to their low toxicity. Such dietary based phytochemicals are used to treat several diseases including cancer. Among these safer phytochemicals, carotenoids belong to the class of tetraterpenes that impart chemical reactivity and photochemical properties to plants. Multiple carotenoids such as lycopene, lutein, astaxanthin, β-carotene, β-cryptoxanthin, crocin, zeaxanthin, and fucoxanthin were explored in various cancers that resulted in significant anticancer efficacy following different signaling pathways. Specifically, carotenoids were also evaluated in blood cancer including leukemia, lymphoma, and myeloma. It was revealed that chemotherapy failure in blood cancer due to poor response to apoptosis resistance and inductive autophagy was significantly improved through the use of carotenoids. Multiple research studies on carotenoids have shown that the therapeutic potential of carotenoids in blood cancer is attributed to various mechanisms such as activation of AMPK, expression of autophagy biochemical markers (p62/LC3-II), activation of Keap1-Nrf2/EpRE/ARE signaling pathway, nuclear factor kappa-light-chain-enhancer of activated B cells (NF-κB), increased level of reactive oxygen species, cleaved poly (ADP-ribose) polymerase (c-PARP), c-caspase-3, -7, decreased level of Bcl-xL, cycle arrest at the G0/G1 phase and decreasing STAT3 expression. The targeting of mentioned signaling pathways and mechanisms has opened new avenues in blood cancer therapy however need further exploration in the arena of blood cancer to overcome barriers in blood cancer therapy. The potential preclinical studies suggested that these molecules needed to be explored in clinical settings for possible new effective therapeutics.

## 8. Conclusions

It is concluded that carotenoids are a versatile class of phytochemicals which have been explored for targeting various cancers. However, in the arena of blood cancer it has been evaluated as an alternative therapy approach due to its potential of overcoming the challenges associated with blood cancer treatment particularly chemotherapy. Carotenoids through various signaling pathways and mechanism such as activation of AMPK, expression of autophagy biochemical markers (p62/LC3-II), activation of Keap1-Nrf2/EpRE/ARE signaling pathway, nuclear factor kappa-light-chain-enhancer of activated B cells (NF-κB), increased level of reactive oxygen species, cleaved poly (ADP-ribose) polymerase (c-PARP), c-caspase-3, -7, decreased level of Bcl-xL, cycle arrest at the G0/G1 phase and decreasing STAT3 expression results in apoptosis induction and inhibition of cancer cell proliferation. It best explains the mechanistic insights behind the particular blood cancer type therapy. It is to worth mention that the literature on carotenoids targeting blood cancer is still not enough and needs more exploration to find out more and new horizons in the blood cancer therapy for the improvement of blood cancer patient life quality. Furthermore, clinical trials on carotenoids significantly could left up it from bench top to the markets.

## Figures and Tables

**Figure 1 nutrients-14-01949-f001:**
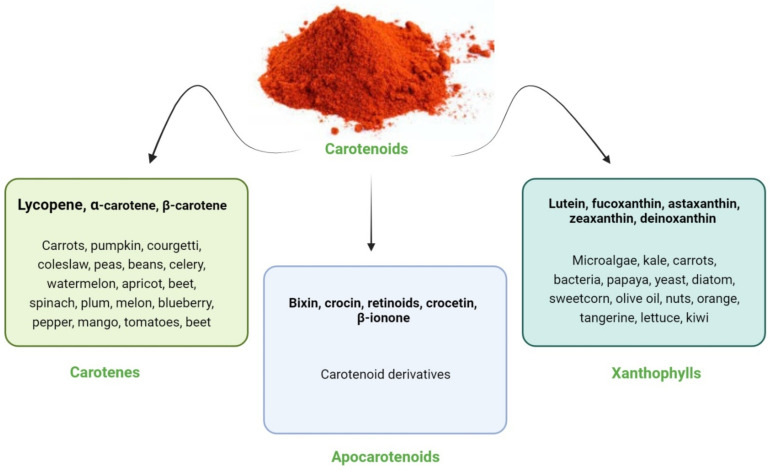
Classification and sources of carotenoids.

**Figure 2 nutrients-14-01949-f002:**
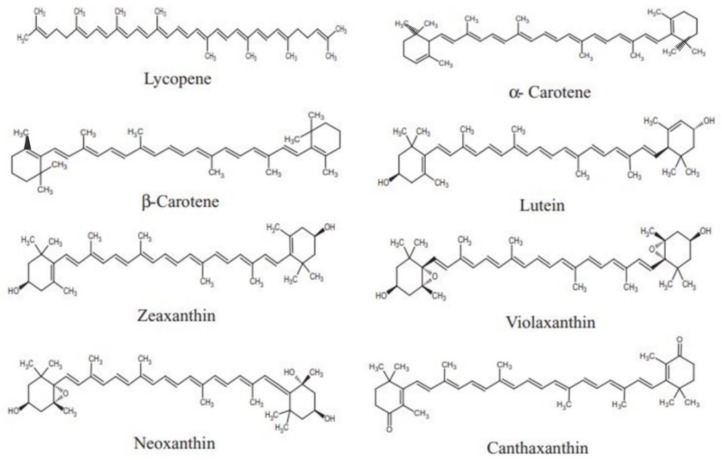
Structures of carotenoids.

**Figure 3 nutrients-14-01949-f003:**
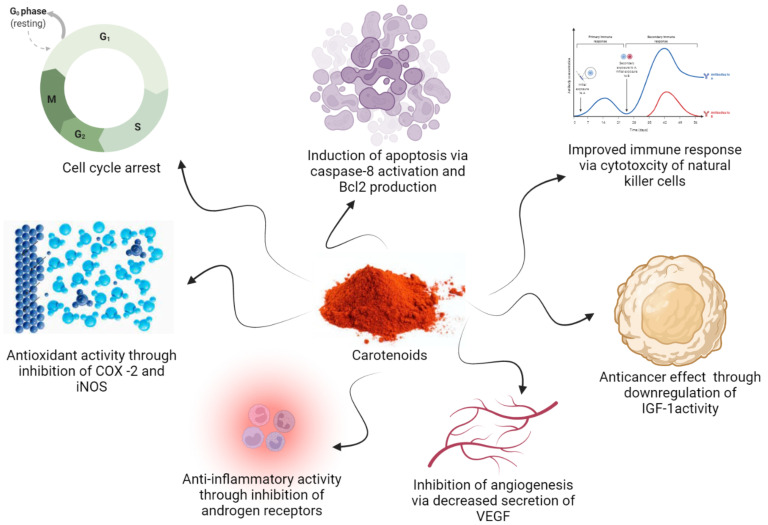
Molecular Mechanism for bioactivities of Carotenoids.

**Figure 4 nutrients-14-01949-f004:**
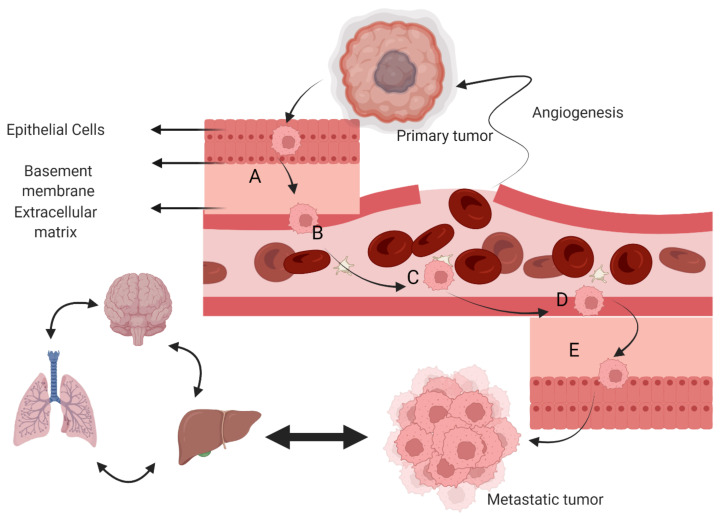
Various involved steps in metastasis: (**A**) local invasion, (**B**) Intravasation, (**C**) circulation in blood or lymphatic vessels, (**D**) extravasations, and (**E**) colonization and metastasis. Infiltration of cancer cells from a primary tumor to surrounding parenchyma take place and traverse through the blood vessel’s membrane.

**Table 1 nutrients-14-01949-t001:** Major representative of dietary carotenoids.

Carotenoid	Diet Source	Reference
β-Cryptoxanthin	Pepper, papaya, oranges, tangerine	[75]
Lycopene	Water melon, tomatoes, pumpkin	[76,77]
α-Carotene	Carrots, green leafy vegetables, coleslaw, pumpkin	[78,79]
Lutein/zeaxanthin	Cucumber, pumpkin, celery, broccoli, spinach, egg, beans, pepper, grapes, melon, carrots, beans	[80,81]

**Table 2 nutrients-14-01949-t002:** Carotenoids and their anticancer potential.

Carotenoids	Cancer Type	Study Design	Mechanism	References
Crocin	Breast cancer	4T1 mammary carcinoma cells injected to BALB/c mice	Inhibition of Wnt/β-catenin target genes	[94]
Retinoic acid	Colon cancer	CT26 murine colon cancer cells	Inhibition of nuclear factor-κB, vimentin, β-catenin and increased level of E-cadherin, gap junctions	[95]
Zeaxanthin	Uveal melanoma	C918 cultured uveal melanoma cells	Decreased matrix metalloproteinase, invasion and migration	[96]
β-cryptoxanthin	Gastric cancer	SGC-7901 gastric cancer cells	Apoptosis induction, reduction in AMP-activated protein kinases, and matrix metalloproteinase	[11]
Fucoxanthin	Lung cancer	murine PC9 xenograft, A549 lung cancer cells	Inhibition of Snail family of zinc-finger transcription factors 1, fibronectin and increased level of tissue inhibitors of metalloproteinase	[93]
	Ovarian cancer	SKOV3 ovarian cancer cells	Decreased β-catenin, vimentin and vascular endothelial growth factor	[97]
Lycopene	Oral cancer	Murine CAL-27 oral cancer xenograft	Inhibition of migration and N-cadherin with elevation in E-cadherin	[98]
Lutein	Breast cancer	Human breast cancer cells (MCF-7, MDA-MB-468)	increased ROS generation, activation of p53 signaling, and increased HSP60 expression	[89]
β-carotene	Leukemia	U 937, HL-60 cell line	Antioxidant, apoptosis, Cell cycle arrest	[99]

## Data Availability

Not applicable.

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
