# Peer review of "Therapeutic Role of Carotenoids in Blood Cancer: Mechanistic Insights and Therapeutic Potential"

_nutrients, 2022, doi:10.3390/nu14091949_

Round 1
Reviewer 1 Report
In proportions, the general description of cancer is more than that of blood cancer.
line 70: carotenes instead of carotenenes
Figure 1: what is sweetcaron ? may be sweetcorn?
carrots contain xanthophylls in negligible amounts Figure 2: why beta-carotene is depicted in a different style than the other compoundsThe structure of alfa-carotene and neoxanthin is wrong. Correctly, in alfa-carotene the beta ring is situated in left side, and alfa-ring in right side.
Similarly in the neoxanthin, the allene end is situated in left side and epoxy-end group in right side.
Line 273: Isolancifolide is not a carotenoid
Table 2: Fenretinide and retinamide are synthetic compounds, and not naturally
missing lutein in Table 2
Author Response
S. No. |
Query |
Response by authors |
Reviewer 1 |
||
1. |
In proportions, the general description of cancer is more than that of blood cancer. |
Actually, the purpose of in-depth details of carious cancer is to provide a brief view to the readers in order to ensure proper background toward blood cancer. |
2. |
Line 70: carotenes instead of carotenenes |
Rectified |
3. |
Figure 1: what is sweetcaron ? may be sweetcorn? |
It is sweetcorn and was rectified |
4. |
Figure 2: why beta-carotene is depicted in a different style than the other compounds. The structure of alfa-carotene and neoxanthin is wrong. Correctly, in alfa-carotene the beta ring is situated in left side, and alfa-ring in right side. Similarly in the neoxanthin, the allene end is situated in left side and epoxy-end group in right side |
Rectified accordingly |
5. |
Line 273: Isolancifolide is not a carotenoid
|
Yes, it is not carotenoid, but was wrongly written in the reference. Therefore, Removed |
6. |
Table 2: Fenretinide and retinamide are synthetic compounds, and not naturally |
Rectified accordingly |
7. |
Missing lutein in Table 2 |
Rectified accordingly |
Reviewer 2 Report
The manuscript "Therapeutic role of Carotenoids in blood cancer: Mechanistic insights and therapeutic potential" summarizes the therapeutic use of natural carotenoids for blood cancer.
While in parts the manuscript goes into unnecessary depth, ie. for the description of the different types of blood tumors which can be found in a medical handbook or the other treatments for blood tumors. however, information such as success rate are not included. On more important topics information is missing or the information provided is superficial. The fact that blood tumors are considered metastatic tumors (Line125-132) needs to be explained in more detail citing e.g. 10.1038/s41568-021-00355-z. Usually, blood tumors in the traditional way are considered to be not metastasize.
L. 143 should be "selective cancer cell ..."
L.144-145: it should be mentioned how successful CAR-T is and in which patients
L. 149-150 statement not clear
Figure 1. Inconsistent! What is the natural source of bixin, crocetin,...?
L. 174-175. statement not clear
The introduction is far too long. The data presented for the efficacy of carotenoids are mainly in vitro data without the distinction of healthy cell response to tumor cell line response. Moreover, the authors do not reflect on the feasibility of the treatment (if the high concentration in vitro show efficacy how does that translate into quantities to be taken by the patient?).
That carotenoids prevent the production of ROS is a known fact but how this relates to the quantities found in vitro and the low hydrophilicity of the compounds and the problem of how to administer the amounts of carotenoids found in vitro is not clear from this article and is not even discussed.
Author Response
1. |
While in parts the manuscript goes into unnecessary depth, ie. for the description of the different types of blood tumors which can be found in a medical handbook or the other treatments for blood tumors. however, information such as success rate are not included. |
Actually, the purpose of in depth details of carious cancer is to provide a brief view to the readers in order to ensure proper background toward blood cancer. In addition, the success rate is not fully understood due to lack of clinical trials in this domain. That’s why it is out of scope of this article. |
2. |
On more important topics information is missing or the information provided is superficial. The fact that blood tumors are considered metastatic tumors (Line125-132) needs to be explained in more detail citing e.g. 10.1038/s41568-021-00355-z. Usually, blood tumors in the traditional way are considered to be not metastasize |
Rectified accordingly |
3. |
L. 143 should be "selective cancer cell ..." |
Rectified accordingly |
4. |
L.144-145: it should be mentioned how successful CAR-T is and in which patients |
Rectified accordingly |
5. |
L. 149-150 statement not clear |
The statement was made clear |
6. |
Figure 1. Inconsistent! What is the natural source of bixin, crocetin,..? |
Crocin and crocetin are found in saffron flowers, and Gardenia fruits, and bixin in Bixa orellana. While β-ionone is found in Lawsonia inermis and Boronia megastigma. |
7. |
L. 174-175. statement not clear |
The sentence was made clear |
8. |
The introduction is far too long. The data presented for the efficacy of carotenoids are mainly in vitro data without the distinction of healthy cell response to tumor cell line response. Moreover, the authors do not reflect on the feasibility of the treatment (if the high concentration in vitro show efficacy how does that translate into quantities to be taken by the patient?). |
The needful changes are made in the introduction. This is a review article based on the analysis of available data and we cannot suggest treatment as no data reported yet.
|
9. |
That carotenoids prevent the production of ROS is a known fact but how this relates to the quantities found in vitro and the low hydrophilicity of the compounds and the problem of how to administer the amounts of carotenoids found in vitro is not clear from this article and is not even discussed. |
In this review, we have discussed the available data and we are limited to that as well. We cannot make any decision/opinion without scientific data. |
Round 2
Reviewer 1 Report
Figure 2.: I still don’t understand why the formula for alpha- and beta-carotene is drawn in a different style than the others. Please draw similar to what is zeaxanthin and lutein.
Table 2.: Curcumin is not carotenoid. Please delete it.
Author Response
Dear Editor/reviewer
Many thanks for your time and queries. We have answered the queries as mentioned against each in the Table below. The revised version of the manuscript is in “Track Changes” mode.
S. No. |
Query |
Response by authors |
Reviewer 2 |
||
1 |
Figure 2.: I still don’t understand why the formula for alpha- and beta-carotene is drawn in a different style than the others. Please draw similar to what is zeaxanthin and lutein. |
Rectified accordingly |
2 |
Table 2.: Curcumin is not carotenoid. Please delete it.
|
Deleted |